# Towards fully covariant machine learning

**Soledad Villar**[*]                                                   *soledad.villar@jhu.edu*
*Department of Applied Mathematics and Statistics, Johns Hopkins University*
*Mathematical Institute for Data Science, Johns Hopkins University*
*Flatiron Institute, a division of the Simons Foundation*

**David W. Hogg**[*]                                                   *david.hogg@nyu.edu*
*Center for Cosmology and Particle Physics, Department of Physics, New York University*
*Max Planck Institute for Astronomy, Heidelberg*
*Flatiron Institute, a division of the Simons Foundation*

**Weichi Yao**                                                        *weichiy@umich.edu*
*Michigan Institute for Data Science, University of Michigan*

**George A. Kevrekidis**                                              *gkevrek1@jhu.edu*
*Department of Applied Mathematics and Statistics, Johns Hopkins University*
*Los Alamos National Laboratory*

**Bernhard Schölkopf**                                                *bs@tuebingen.mpg.de*
*Max Planck Institute for Intelligent Systems and ELLIS Institute, Tübingen*

**Reviewed on OpenReview:** *https://openreview.net/forum?id=gllUnpYuXg*

## Abstract

Any representation of data involves arbitrary investigator choices. Because those choices are external to the data-generating process, each choice leads to an exact symmetry, corresponding to the group of transformations that takes one possible representation to another. These are the *passive symmetries*; they include coordinate freedom, gauge symmetry, and units covariance, all of which have led to important results in physics. In machine learning, the most visible passive symmetry is the relabeling or permutation symmetry of graphs. The *active symmetries* are those that must be established by observation and experiment. They include, for instance, translations invariances or rotation invariances of physical law. These symmetries are the subject of most of the equivariant machine learning literature. Our goal, in this conceptual contribution, is to understand the implications for machine learning of the many passive and active symmetries in play. We discuss *dos and don'ts* for machine learning practice if passive symmetries are to be respected. We discuss links to causal modeling and argue that the implementation of passive symmetries is particularly valuable when the goal of the learning problem is to generalize out of sample. We conjecture that the implementation of passive symmetries might help machine learning in the same ways that it transformed physics in the twentieth century.

## 1 Introduction

Many important ideas in machine learning (ML) have come from—or been inspired by—mathematical physics. These include the kernel trick (Courant & Hilbert, 1953; Schölkopf & Smola, 2002) and the use of statistical mechanics techniques to solve probabilistic problems (Hastings, 1970; Gelfand, 2000). Here we suggest another connection between physics and ML, which relates to the representation of observables: When features and labels are represented in a mathematical form that involves investigator choices, methods of ML (or any

---

[*]joint first author

relevant model, relationship, method, or function) ought to be written in a form that is exactly equivariant to changes in those investigator choices. These ideas first appear in the physics literature in the 1910s (most famously in Einstein 1915). They are given in the introduction of *The Classical Groups* (Weyl, 1946) as a motivation to study group theory. Literally the first sentences of *Modern Classical Physics* (Thorne & Blandford, 2017) are

> [...] a central theme will be a Geometric Principle: The laws of physics must all be expressible as geometric (coordinate-independent and reference-frame-independent) relationships between geometric objects (scalars, vectors, tensors, ...) that represent physical entities.

This Geometric Principle leads to the important physical symmetries of coordinate freedom and gauge symmetry; a small generalization would include what we will refer to as units covariance (see the glossary in Appendix A). Each of these symmetries has led to fundamental results in physics. Some of these ideas are also exploited in ML, in particular in the geometric deep learning literature (Bronstein et al., 2021; Weiler et al., 2021a). We argue—in this purely conceptual contribution—that analogs of these symmetries could have an impact on ML, and thus increase the scope of group-equivariant methods in ML.

In natural science, there are two types of symmetries (see, for example, Section 4.1 of Rovelli & Gaul 2000). The first kind is *passive*, arising from the arbitrariness of the mathematical representation described above. An example familiar in ML is the equivariance of functions on graphs to the relabeling of the graph nodes. This is an exact, passive symmetry; graph neural network architectures (GNNs) build this passive symmetry in by design (Bruna et al., 2013; Duvenaud et al., 2015; Gilmer et al., 2017; Hamilton, 2020). See Huang et al. (2023) for a recent work describing passive and active symmetries in GNNs. An example familiar to physicists is what we call *units covariance*, which is the requirement that any correct description of the world has inputs and outputs with the correct units (and dimensions; see Appendix A for definitions). This symmetry is extremely powerful (see Section 3).

The second kind of symmetries is *active*. These are the ones that must be established by observations and experiments. The fundamental laws of physics do not seem (at current precision) to depend on position, orientation, or time, which in turn imply conservation of momentum, angular momentum, and energy (the celebrated theorem of Noether 1918). Active symmetries like these are empirical and could (in principle) be falsified by experimental tests. Both active and passive symmetries can be expressed in terms of group or groupoid actions and equivariances, but their epistemological content and range of applicability are very different.

In this contribution, we argue that passive symmetries apply to essentially all data analysis problems. They have implications for how we structure ML methods. Although we provide some examples, most of our contributions are conceptual.

**Our contributions:**

- We introduce the concept of passive and active symmetries to ML. We give a formal definition of passive and active symmetries in terms of group actions and explain how passive symmetries are always in play in problems using real-world data.

- We provide guidance on how to structure ML models so that they respect the passive symmetries. We call out some current standard practices that can prevent models from obeying symmetries. We give particularly detailed guidance in the context of data normalization.

- We illustrate with toy examples how enforcing passive symmetries in model structure and in normalization can improve regressions.

- We demonstrate that imposing passive symmetries can lead to the discovery of important hidden objects in a data problem. We show that a passively equivariant problem can often be made actively equivariant by the introduction of latent variables with appropriate group-theoretic properties.

- We draw connections with causal inference. One is that all causal graphs and mechanistic models are constrained to be consistent with the passive symmetries. Another is that the determination that a data problem has all the inputs necessary to express the symmetry exactly looks like a causal inference. We also explain how active symmetries can be expressed in terms of interventions.

- We provide a glossary (in Appendix A) that can be used to translate terminology and relate ideas between physics and ML.

## 2 Passive symmetries

Passive symmetries arise from redundancies or free parameters or investigator choices in the representation of data. They are to be contrasted with the active symmetries, which arise from observed or empirical invariances of the laws of physics with respect to parameters, like position, velocity, particle labeling, or angle. Passive symmetries can be established with no need of observations, as they arise solely from the principle that the physical world is independent of the mathematical choices we make to describe it. The groups involved in coordinate freedom can be large and complicated (for example, groups of reparameterizations).

In contrast, a big part of the literature on equivariant ML is implicitly or explicitly looking at *active* symmetries. This is possibly because in most problems the coordinate system is fixed before the problem is posed, and both training and test data are expressed in those fixed coordinates. If a data set is made with a fixed coordinate system, but still exhibits an *observable* invariance or equivariance with respect to (say) rotations, then that represents an active symmetry. However, cases of exact active symmetries are rare; they only really appear in natural-science contexts like protein folding or cosmology. For example, in a protein folding problem, the way the protein folds may not depend on its orientation in space (rendering the problem actively $O(3)$ equivariant). This finding relies on the (empirical) observation that the local gravitational field (on Earth), for example, does not affect the folding. This may be approximately true or assumed or experimentally established; it is an active symmetry. In contrast, the fact that the protein folds in a way that doesn't depend on the coordinate system *chosen to describe it* is absolutely and always true; it is not experimentally established; it is a passive symmetry.

The relationship between active and passive symmetries is reflected in the relationship between what are sometimes called active and passive transformations, or *alibi* and *alias* transformations, depicted in Figure 1. An active or alibi transformation is one in which the objects of study are moved (rotated, translated, interchanged, etc.). A passive or alias transformation is one in which the coordinate system in which the objects are represented is changed (rotated, translated, relabeled, etc.). Mathematically, the two seem very similar: For example, how do we know whether we rotated all the vectors in our problem by 30 deg, or else rotated the coordinate system used by $-30$ deg? The answer is that if you rotated *absolutely all* the vectors (and tensors) in your problem, including possibly many latent physical vectors, then there would be no mathematical difference. However, this is not possible in practice. In real problems, where some vectors can't be actively rotated (think, for example of the local gravitational-field vector, or the vector pointing towards the Sun), or some may not be known or measurable, the two kinds of transformations are different.

This problem—that a passive symmetry only becomes an active symmetry when it is possible to transform every relevant thing correspondingly—suggests that it might be hard to implement or enforce an exact passive symmetry in a real data-analysis problem. It requires us to incorporate all relevant contextual information. How do we know if all relevant features are part of our data set? We could perform the protein-folding experiment in a closed, isolated environment to make sure no external forces are in play? This is impossible for many practical applications, and furthermore, there could still exist fundamental constants that are not part of our model or knowledge (see Section 5). Another approach is to perform the experiment multiple times after actively putting the molecules into different orientations. If the protein folds differently, we learn that the problem is not symmetric with respect to the 3d coordinates of the molecule, and therefore when a rotation is performed there must be at least one more vector that needs to be rotated as well (for instance, the gravity vector or the eigenvectors of some stress tensor, say). This identification of all necessary inputs to establish the passive symmetry is similar to the problem of performing interventions to learn the existence of confounding factors in causal inference (see Section 6).

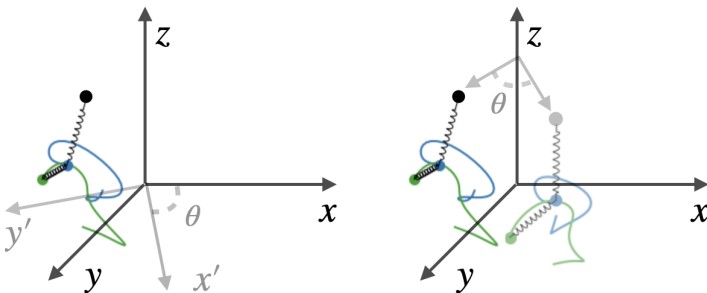

Figure 1: Figure depicting the difference between active and passive transformations. *(Left panel)*: The passive or alias transformation corresponding to a rotation of the coordinates through an angle $\theta$ in the xy-plane. The equivariance of the dynamics with respect to this transformation is a passive symmetry. *(Right panel)*: The active or alibi transformation corresponding to a rotation of the double pendulum state through an angle $-\theta$ in the xy-plane. Equivariance with respect to this transformation is an active symmetry.

Once a passive symmetry—and all relevant contextual information—is identified, we want to write the data analysis problem or learned function such that it is exactly *equivariant* with respect to the relevant group: If the coordinate system of the inputs is changed, the coordinate system of the output should change correspondingly. ML methods that are not constrained to respect passive symmetries are doomed to make certain kinds of mistakes. We will provide some examples in Section 8 and Section 9.

The most restrictive form of the Geometric Principle quoted in Section 1 states that physical law must be written in terms of vectors, tensors, and (coordinate-invariant) scalars. These objects can only be combined by rules set out in the Ricci calculus (Ricci & Levi-Civita 1900; sometimes called Einstein summation notation, Einstein 1916). This calculus was introduced to make objects equivariant to coordinate diffeomorphisms on curved manifolds, but it applies to $O(3)$ and Lorentz symmetry as well. In the Ricci calculus, objects are written in index notation (a scalar has no indices, a vector has one index, and a $k$-tensor has $k$ indices), outer products are formed, and only certain kinds of sums over pairs of indices are permitted. When the inputs to a function are scalars, vectors, and tensors, and the function conforms to the rules of the Ricci calculus, the function will produce a geometric output (a scalar, vector, or tensor,[0] depending on the number of unsummed indices), and the function will be precisely covariant to rotations and reflections of the coordinate system. This is how a large class of passive symmetries is enforced in physics contexts.

There are many other passive symmetries, including coordinate diffeomorphisms, reparameterizations (including canonical transformations in Lagrangian and Hamiltonian systems), units covariance (see Section 3), and gauge freedom. Some are easy to implement in ML contexts and some are difficult; not all group equivariances have practical implementations useful for ML methods available at present. Convolutional neural networks (LeCun et al. 1989) are very effective even when strict translation symmetry is not active; this effectiveness might be related to the connection to the passive translation symmetry of the image pixel grid.

Sometimes it is difficult to tell whether a symmetry is active or passive. For example, the law of gravity is explicitly $O(3)$ invariant: If you rotate all of the position vectors, you rotate all of the forces correspondingly too. If you rotate all of the initial conditions, you rotate all of the gravitational trajectories correspondingly too. But this symmetry is also a passive symmetry: The law of gravity does not depend on the orientation (or handedness) of the coordinate system.

If an active symmetry is in play, the physical law can often be written in terms of invariants such that the symmetry is enforced by construction or definition, thereby making it also a passive symmetry. The discovery of general relativity (Einstein, 1915) can be seen as the discovery that active symmetries—the speed of light is the same for all observers, and the gravitational mass equals the inertial mass—can be converted

---

[0]It should be noted here that with the word "vector" and "tensor" here we are making specific technical reference to true vectors and tensors in 3-space, subject to passive $O(3)$ symmetries, like (physical) velocities, accelerations, and stress tensors. We are *not* including arbitrary lists or tables of data or coefficients, which are sometimes called "vectors" and "tensors" in ML contexts. See Appendix A for more detail.

into passive symmetries: When laws are written in a form consistent with the passive symmetries, they automatically enforce the corresponding active symmetries. Indeed, the equations of general relativity were found by looking at every differential equation (to some degree) consistent with the passive symmetry of coordinate diffeomorphisms on curved spacetime manifolds until one was found that reduced to Newtonian gravity in the weak-field limit.

We stress that these kinds of insights had a big impact on physics (Earman & Glymour, 1978); only a few years prior to 1915, such considerations would have seemed highly unusual; now this form of argument is canon in theoretical physics (see, for example, Zee 2016). That evolution motivates this contribution; passive symmetries do not feature in most of today's ML practice—if the development of physics is any indication, their potential could be significant. Some valuable ML research has started recently along these directions (Weiler et al., 2021a; Bronstein et al., 2021).

## 3 Example: Units covariance

Perhaps the most universal passive symmetry is units covariance—the behavior of a system does not depend on the units system in which we write the measured quantities. It is a passive symmetry with extremely useful consequences.

Consider a mass $m$ near the surface of the Earth, close enough to the surface such that the gravitational field can be considered to be determined by a constant (not spatially varying) vector with magnitude $g$ and direction downwards. *Question 1:* If this mass $m$ is dropped (released at rest) from a height $h$ from above the ground, how much time $T$ does it take to fall to the ground? *Question 2:* If this mass $m$ is launched from the surface at a velocity of magnitude $v$ at an angle $\theta$ to the horizontal, how much horizontal distance $L$ will it fly before it hits the surface again? Assume that only $m, g, h$ enters the solution;[1] In particular, assume that the height $h$ and the velocity $v$ are both small enough so that air resistance, say, can be ignored.

The answers to these questions are almost completely determined by dimensional (or units-covariance) arguments (see Appendix A for definitions). The mass $m$ has units of kg, the gravitational acceleration magnitude $g$ has units of $\mathrm{m\,s^{-2}}$, the velocity magnitude $v$ has units of $\mathrm{m\,s^{-1}}$, the time $T$ has units of s, and the lengths $h$ and $L$ have units of m. The angle $\theta$ is dimensionless. The only possible combination of $m, g, h$ that has units of time is $\alpha\sqrt{h/g}$, where $\alpha$ is a dimensionless constant, which doesn't depend on any of the inputs. The only possible combination of $m, g, v, \theta$ that has units of length is $\beta(\theta)\,v^2/g$, where $\beta(\theta)$ is a dimensionless function of a single dimensionless input. That is, both Question 1 and Question 2 can be answered up to a dimensionless prefactor without any training data. And both of those answers don't depend in any way on the input mass $m$ (which is the fundamental observation that leads to general relativity; Einstein 1915).

This shows that a function—and even a fundamental physical relationship—can sometimes be inferred from units covariance only, that is, from a purely passive symmetry, combined with the *empirical* knowledge that the solution is *independent* of all other observables (which itself is a causal assumption). Unit covariance is widely used in numerical methods, and it has been discussed in ML previously (Villar et al., 2023; Bakarji et al., 2022; Xie et al., 2022). It can help with training, predictive accuracy, and out-of-sample generalization.

## 4 Formal definition

Consider $\mathscr{X}$ to be space of all possible physical states of a specific system (for instance $x \in \mathscr{X}$ could be the positions, velocities, masses, spins, and charges of a set of particles, at a time or possibly at a set of times). We consider maps $\{\Phi_i : \mathscr{X} \to \mathscr{H}\}_{i \in \mathscr{I}}$ where $\mathscr{H}$ is the space of encodings (or representations) of the values of those positions, velocities, masses, and so on, in some units system and some coordinate system. That is, any element $z \in \mathscr{H}$ will be a list of real values of vector and tensor components and scalars. Different maps $\Phi_i$ will have (in general) different coordinate origins, different axis orientations, and different units of measurement.

---

[1]We will return to this seemingly innocuous point below. We note that it is an *empirical* statement, which will turn out to have rather significant implications.

Provided that every $\Phi_i$ records all of the information necessary to describe the state $x \in \mathscr{X}$ (or, equivalently, every element $z \in \mathscr{H}$ contains all of the information necessary to describe the state), for any two encodings $\Phi_i$ and $\Phi_j$ there is an invertible morphism $\beta_j^i : \mathscr{H} \to \mathscr{H}$ that makes the diagram (1) commute.

$$
\begin{array}{ccc}
\mathscr{X} & \xrightarrow{\;id\;} & \mathscr{X} \\
{\scriptstyle \Phi_i}\downarrow & {\scriptstyle \beta_j^i} & \downarrow{\scriptstyle \Phi_j} \\
\mathscr{H} & \xrightarrow{\;\;\;\;\;} & \mathscr{H}
\end{array}
\tag{1}
$$

The passive symmetries comprise the group $P$ of invertible morphisms $\beta_j^i$ that makes the diagram commute. The group $P$ consists of all the possible changes of units or coordinates or automorphisms between encodings or observables.

For example, take $\mathscr{X}$ to be the space of states of a protein molecule. Each map $\Phi_i$ could encode the positions of each of its atoms in some coordinate system. The passive symmetries include reordering of the atoms, changes of the coordinate system by any invertible morphism, and changes to the units in which positions (lengths) are measured.

Active symmetries, on the other hand, can be thought of as transformations of the world that preserve an observable property. They involve interventions in the physical system, and therefore they are typically empirical and approximate. Not every passive symmetry corresponds to an active symmetry, nor vice versa.

To define the active symmetries we fix $\Phi : \mathscr{X} \to \mathscr{H}$ an encoding as above; a function $F : \mathscr{H} \to \mathscr{Y}$, where $F$ is a function that delivers a possible observable or prediction or system property; and a group $G$ that acts on the $\mathscr{X}, \mathscr{H}$ and $\mathscr{Y}$ via actions $\tau$, $\beta$ and $\rho$ respectively. An element $y \in \mathscr{Y}$ contains some prediction or observable of interest in the system, such as the energy or the future time evolution (trajectory). Given a group element $g \in G$ we might draw this commutative diagram:

$$
\begin{array}{ccccc}
\mathscr{X} & \xrightarrow{\;\Phi\;} & \mathscr{H} & \xrightarrow{\;F\;} & \mathscr{Y} \\
{\scriptstyle \tau(g)}\downarrow & & {\scriptstyle \beta(g)}\downarrow & & \downarrow{\scriptstyle \rho(g)} \\
\mathscr{X} & \xrightarrow{\;\Phi\;} & \mathscr{H} & \xrightarrow{\;F\;} & \mathscr{Y}
\end{array}
\tag{2}
$$

We say that the tuple $(G, F, \beta, \rho)$ represents an active symmetry of $\mathscr{X}$ if the diagram (2) commutes.

To continue the example in which $\mathscr{X}$ is the space of states of a protein molecule: If $F$ computes the total electrostatic energy of the protein, there is an active symmetry corresponding to the group $G = O(3)$ of rotations and reflections, in which $\beta(g)$ is the standard matrix representation of $g$ applied to all the position vectors, and $\rho(g)$ is the identity operator for all $g \in G$. This symmetry is active in the sense that it says how something changes (the energy, and it doesn't) when the molecule's state is changed (it is rotated in space).

The alias vs alibi distinction (Section 2) maps onto the correspondence of related passive and active symmetries. The active symmetries correspond to *interventions* in the system, while passive symmetries are purely in the realm of representation. Most of the equivariant ML literature is focused on active symmetries. Projects begin with the question: What active symmetries are in play in this system? In what follows, we reserve the word "equivariance" for active symmetries and use the word "covariance" for passive symmetries, consistent with the use of the word "covariance" in physics contexts (see Appendix A).

## 5 Experiments and examples

**Black-body radiation:** An important moment in the history of physics was the discovery that the electromagnetic radiation intensity $B_\lambda$ (energy per time per area per solid angle per wavelength) of thermal black-body radiation can be described with a simple equation (Planck, 1901)

$$
B_\lambda(\lambda) = \frac{2 \, h \, c^2}{\lambda^5} \left[ \exp \frac{h \, c}{\lambda \, k \, T} - 1 \right]^{-1} ,
\tag{3}
$$

where $h$ is Planck's constant, $c$ is the speed of light, $\lambda$ is the wavelength of the electromagnetic radiation, $k$ is Boltzmann's constant, and $T$ is the temperature. In finding this formula, Planck had to posit the existence (and units) of the constant $h = 6.62607015 \times 10^{-34}\,\mathrm{kg\,m^2\,s^{-1}}$ (Planck's original value was presented with less precision and in erg s, which are different units but the same dimensions). Prior to the introduction of $h$, the only dimensionally acceptable expression for the black-body radiation intensity was $B_\lambda(\lambda) = 2\,c\,k\,T/\lambda^4$, which is the long-wavelength (infrared) or high-temperature limit of (3). Planck's discovery solved the "ultraviolet catastrophe" of classical physics. This is the problem that, classically, the black-body spectrum, or any thermal object, ought to contain infinite numbers of excited modes at short wavelengths, or high frequencies, and thus infinite energy density. Planck's solution seeded the development of quantum mechanics, which governs the behavior of all matter at small scales, and which cuts off the ultraviolet modes through quantization of energy.

Planck's problem can be solved almost directly with the passive symmetry of units covariance. That is, the exponential cut-off of the intensity appears at a wavelength set by the temperature and a new constant, that must have units of action (or action times $c$, or action divided by $k$, or one equivalent in terms of the lattice of dimensional features, see Appendix B and Villar et al. 2023).

In Figure 2 (left) we perform the following toy experiment: We generate noisy samples of intensities as a function of wavelength and temperature according to (3), and the learning task is to predict the intensity for different values of wavelengths and temperatures. We perform three experiments, described in Appendix B *(A)* a units-covariant regression (employing the approach of Villar et al. 2023) using only $\lambda, T, c, k$; *(B)* a units covariant regression with an extra dimensional constant found by cross-validation; and *(C)* a standard multi-layer perceptron regression (MLP) with no units constraints. Our results show that no units-covariant regression for the intensity as a function of $\lambda, T, c, k$ can reproduce accurately the intensity $B_\lambda$. However when the regression is permitted to introduce a new dimensional constant (but enforce exact units-covariance given the new constant), it finds a constant with units that is consistent with $h$ (or $h$ times a combination of $c$ and $k$). The units-covariant model with an extra constant outperforms the baseline MLP. Naïvely this suggests that passive symmetry brings new capabilities.

**Springy double pendulum:**   The double pendulum connected by springs is a toy example often used in equivariant ML demonstrations (Finzi et al., 2021; Yao et al., 2021; Villar et al., 2023). The final conditions (position and velocities of both masses after elapsed time $T$) are related to the initial conditions (position and velocities of the masses at the initial time), and the dynamics is classically chaotic. This means that prediction accuracy, in the end, must be bounded by considerations of the fundamental mathematical properties of dynamical systems.

The system is subject to a passive $O(3)$ symmetry (equivariance with respect to orthogonal coordinate transformations), an active $O(2)$ symmetry (equivariance with respect to rotations and reflections in the 2D plane normal to the gravity), and an active time-translation symmetry, arising from the fact that the total energy is conserved. The $O(3)$ symmetry is passive, because it is guaranteed by the fact that all vectors must be described in a coordinate system; nothing physical can change as the vectors undergo passive transformations because of coordinate-system changes. The $O(2)$ symmetry is active, because it is an experimental fact that if the initial conditions are changed by an active or alibi rotation in the plane perpendicular to gravity, the dynamics and final state rotate accordingly. The $O(2)$ active symmetry corresponds to the set of transformations in $O(3)$ that fix the gravity vector.

The passive $O(3)$ symmetry requires that the coordinates of all relevant vectors are transformed identically, the positions and momenta of both masses and the gravity vector. If the model doesn't have access to all relevant vectors as inputs then the predictions will not necessarily be $O(3)$ equivariant. We perform an experiment in which we predict the dynamics of the double pendulum using $O(3)$-equivariant models. The symmetries are implemented by converting the network inputs (scalars and components of vectors) into invariant scalar quantities according to the Ricci calculus (which explicitly encodes $O(3)$), building the model in the space of the invariant scalars (as per Villar et al. 2021). These $O(3)$-invariant scalars are used to parameterize an $O(3)$-invariant Hamiltonian, and the $O(3)$-equivariant dynamics are predicted by integrating the Hamiltonian using a symplectic integrator, as proposed by Sanchez-Gonzalez et al. (2019) and implemented in Yao et al. (2019). More details are given in Appendix C.

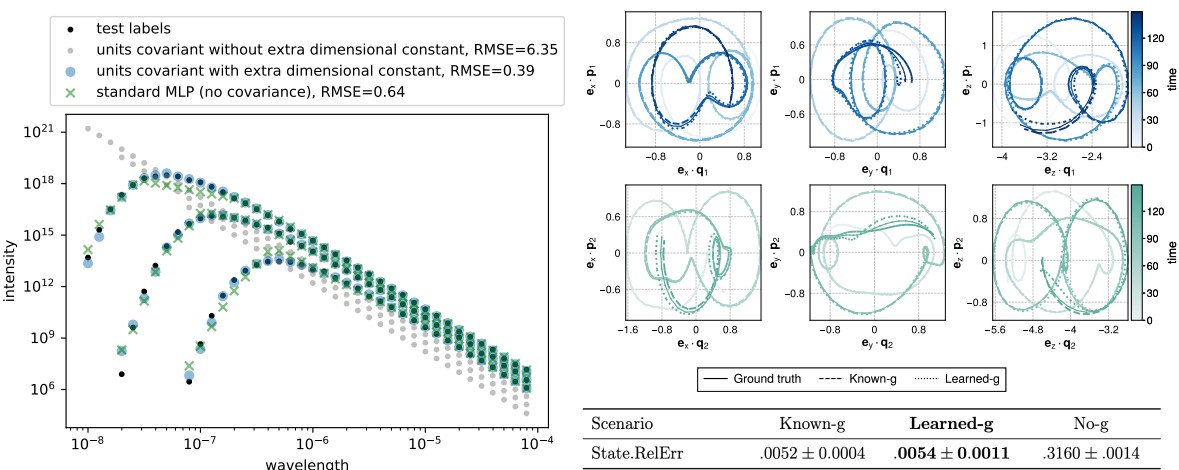

Figure 2: *(Left panel)* We predict the intensity of black body radiation as a function of wavelength and temperature. For all experiments, we use an MLP consisting of 3 layers with 20 hidden units each. The *standard MLP* uses wavelength and temperature as features and it doesn't require the output to be dimensionally correct. The *units covariant without extra constant* learns a scaling of the only dimensionally correct object one can construct with inputs $\lambda, T, c, k$ (see description main in text). The *units covariant with extra dimensional constant* incorporates a constant with units $[\mathrm{kg}, \mathrm{m}, \mathrm{s}, \mathrm{K}] \in \{-2, -1, 0, 1, 2\}^4$ as an input feature, it performs a units covariant regression with the original features $\lambda, T, c, k$, and the extra constant. It then selects a constant with a low validation error and reports the results on the test set. The constant learned for the depicted plot is $1.61\mathrm{e}52$ $\mathrm{kg}^{-1}\mathrm{m}^{-1}\mathrm{s}^{-1}\mathrm{K}^{-1}$, which is the same units and similar magnitude to the valid physical combination $c\,k\,h^{-2}$. *(Right panel)* Performance of learning the dynamics of the springy double pendulum. We consider the three models (described in the main text): (Known-g) an $O(3)$-equivariant model where the gravity is an input to the model, (No-g) an $O(3)$-equivariant model where the gravity is not given, and (learned-g) an $O(3)$-equivariant model that uses the position and momenta as well as an unknown vector that the model learns. The results show that $O(3)$-equivariance permits the learning of the gravity vector from data with only minimal impact on performance. See Appendix C for a more detailed description of the experiment.

The models considered here are *(Known-g)*—an $O(3)$-equivariant model that has the positions, momenta, and gravity vector all as features (similar to the models in Villar et al. 2021; Yao et al. 2021); *(No-g)*—an $O(3)$-equivariant model that is missing the gravity vector as an input feature; and *(Learned-g)*–an $O(3)$-equivariant model that has the position, momenta and an extra unknown vector as features. The latter model optimizes model weights along with the unknown vector. In the right panel of Figure 2 we show the performance of the three models. We remark that in *(Learned-g)*, the learned vector in the performed experiments was nearly parallel to the true (but unknown) gravity vector $g$; the angle between the learned and true gravity vector ended up at 0.00016 radians.

**Non-covariant data normalization**   In section 9 we discuss covariant vs non-covariant data normalization procedures. In appendix C we provide experiments (in the springy double pendulum setting) that empirically show that non-covariant data normalization can hurt the performance significantly, whereas covariant data normalization does not.

# 6   Connections with causality

There is nothing statistical about the notion of passive symmetries, and thus everything we have said above also applies to causal models (Peters et al., 2017). There are, however, a few comments specific to causality.

The passive symmetry discussed in Section 3—and indeed all passive symmetries—can also deliver information pertaining to the (hard problem of) inference of causal structure: treating $g$ as a constant, we can construct a structural causal model with the following vertices: *(a)* an initial value of $v$, *(b)* a value of $m$, chosen

independently, and *(c)* a final value of $L$, affected by a noise term $\theta$. Time ordering implies that possible causal arrows are from $v, m, \theta$ to $L$. As argued above, dimensional analysis rules out the arrow $m \to L$, leaving us with the non-trivial result that in the causal graph, only $v, \theta$ cause $L$. As in Section 3, this conclusion can be reached without any training data or interventions.

That said, dimensional analysis makes a strong assumption, which is that *all* relevant quantities for predicting $L$ have been specified in the list $m, g, v, \theta$. For example, if the projectile is large enough or the speed $v$ is high enough, air resistance will come into play, and the size of the object and the density of air will enter, bringing new variables and new combinations of variables that matter to the answer. This difficulty is related to the problem in causal inference of knowing or specifying all possible confounding variables.

This can also be linked to the notion of experimental interventions. Suppose we assume that only certain quantities come into the solution (say, $m, g, h$). How would we confirm this in practice? In essence, this is not a probabilistic statement, but one about the behavior of a system under interventions. A set of experiments can indicate that a certain outcome (or effect variable) depends on a certain set of input (cause) variables but is independent of certain other potential cause variables. In this case, the physical law is not inferred from dimensional arguments alone, but from a combination of dimensional and causal arguments.

Even if interventions are not available (for $g$, for example), physicists trying to infer a law will not do so based (purely) on input-output data: they will have prior knowledge from related problems informing them as to which variables are relevant. For example, we may know from having previously solved a related problem that we expect a problem to depend on $g$. This is a form of qualitative *transfer* that we expect will also become relevant for model transfer in ML (Rojas-Carulla et al., 2018).

Finally, we remark that causal language appeared above in Section 2 where we implicitly contrast the active symmetries with the passive symmetries in terms of interventions: The active symmetries are those that make predictions for experiments in which interventions have been made (the molecule has been rotated with respect to the gravitational-field vector, for example).

## 7 Connections to current ML practice

Most present-day ML implementations don't impose exact symmetries. Sometimes they approximate equivariances by means of data augmentation (Chen et al., 2020; Huang et al., 2022). In this work we mostly focus on exact symmetries: Given data spaces $X$ and $Y$ and a group $G$ acting on $X$ and $Y$, equivariant ML restricts the function space to those satisfying $f(g \cdot x) = g \cdot f(x)$ for all $f \in \mathscr{F}$, $g \in G$, $x \in X$. There are three main approaches to perform optimization in the space of equivariant functions:

- Parameterizing the space of equivariant functions by extending the notion of group convolutions or weight sharing (Kondor & Trivedi, 2018; Cohen & Welling, 2016; Weiler et al., 2021b).

- Explicitly parameterizing the space of equivariant functions via equivariant linear layers (via irreducible representations (Thomas et al., 2018; Geiger & Smidt, 2022; Kondor, 2018), or otherwise (Maron et al., 2018; Finzi et al., 2020; 2021).

- Finding a set of invariant features and expressing invariant/equivariant functions in terms of those features. Related ideas have been studied in differential geometry and invariant theory (Fels & Olver, 1998), and recently have been used in machine learning (Villar et al., 2021; Blum-Smith & Villar, September 2023; Cahill et al., 2022; Dym & Gortler, 2022; Kaba et al., 2023). This can also be done by imposing symmetry as a soft constraint (Shakerinava et al., 2022; Gupta et al., 2023).

The three approaches achieve similar goals, but their practical implementation may be dramatically different. For example, efficiently computing a complete generating set of invariant/equivariant features may be prohibitive, especially for large finite groups. On the other hand, in some cases, one needs to go to high-order tensors for the linear layers to be expressive enough, and this may be hard to construct. More often, it may be possible to construct such a family, but we may lack proof that they are universal approximators of *all* invariant/equivariant functions within some well-defined context, even in a limiting sense.

Even the non-trivial diffeomorphism symmetries of general relativity have been considered for ML (Weiler et al., 2021a). And aside from the previously mentioned results in Convolutional/Graph Neural Networks, another example of successful exact universal parametrization of a family of functions is the implementation of symplectic networks, which exactly preserve a differential 2-form, the symplectic form, on the associated manifolds (Jin et al., 2020; Burby et al., 2020). Their use most relevant in the study of Hamiltonian systems.

Overall, equivariant ML models can predict the properties and behaviour of physical systems (see Cheng et al. 2019), and have plenty of scientific applications (Batzner et al., 2022; Musaelian et al., 2022; Stärk et al., 2022; Yu et al., 2021; Wang et al., 2022). Interesting theoretical developments analyze the implicit bias, generalization error, and sample complexity of equivariant ML models (Petrache & Trivedi, 2023; Tahmasebi & Jegelka, 2023; Lawrence et al., 2021; Bietti et al., 2021; Elesedy & Zaidi, 2021; Elesedy, 2021; Huang et al., 2023; Mei et al., 2021).

## 8 Dos and Don'ts

MacKay famously wrote (see Muldoon 2021)

> Principal Component Analysis is a dimensionally invalid method that gives people a delusion that they are doing something useful with their data. If you change the units that one of the variables is measured in, it will change all the "principal components"

This comment is aligned with our mission, but also misleading: If a rectangular data set contains only data with identical units (that is, all features of all records have the same units), then PCA does exactly the right thing. That said, if a rectangular data set has features with different units (for example, if every record contains a position, a temperature, a voltage, and a few intensities), then indeed the output of PCA will be extremely sensitive to the units system in which the features are recorded, and thus not invariant to the units choices.

Consider a kernel function with inputs that are lists of features with different units. If the kernel function involves, say, an exponential of a sum of squares of differences of the input features, the output of the kernel function cannot obey the passive symmetry of units covariance. Quantities with different units cannot be summed, and dimensional quantities cannot be exponentiated. On the other hand, if a kernel function can be chosen that is units covariant, then the result of a kernel algorithm can in principle be covariant. These considerations are relevant for the maximum margin hyperplane in kernel support-vector machines (Boser et al., 1992), eigenvectors in kernel PCA (Schölkopf & Smola, 2002), or Gaussian processes (Williams & Rasmussen, 2006).

Learning involves optimization. Optimization is of a scalar cost function. If passive geometric groups are in play, like $O(3)$, the parameters that are explicitly or implicitly components of vectors can only be combined into the scalar objective through the Euclidean norm. Otherwise the scalar objective isn't scalar in the geometric sense of "invariant to $O(3)$", and the optimization won't return a result that is invariant (or equivariant) to $O(3)$. Similarly, if the components of the vector are normalized differently before they are summed in quadrature, the objective won't be invariant to $O(3)$. And similarly, if all the different contributions to the objective aren't converted to the same units before being combined, the model won't be units covariant. The common practices of making objectives with functional forms other than Euclidean norm, normalizing features with data ranges, and combining features with different units, all make common ML methods, by construction, inconsistent with the passive symmetries in play. We say more about normalization below in Section 9.

Neural nets, in their current form, violate many rules. For example: Transcendental functions like `exp()` and `arctanh()` and most other nonlinear functions can only be applied to scalars—that is, not components of vectors or tensors but only scalars—and only dimensionless. That means that the nonlinearities in neural networks are (or should be) implicitly predicated on the weights removing the units of the input features, and the linear combinations performing some kind of dot products on the inputs. That, in turn, means that the internal weights in the bottom and top layers of a neural network *implicitly* have geometric properties and units. They have geometric properties and units such that the latent variables passed into the nonlinear

functions are dimensionless scalars. Because they have these properties, a trained neural network cannot be covariant in the end, unless the inputs and outputs are already covariant scalars.

There are exceptions to the restrictions on nonlinear functions: If nonlinearities are mathematically homogeneous, as it is for a pure monomial, or for the RELU function, dimensional scalars (but not vector or tensor components) can be taken as inputs. It is interesting to ask whether the success of RELU in ML might be related to its homogeneity.

Above we said that the weights in a neural network implicitly have geometric properties and units. What are these? Imagine, say, at the input layer, that three of the inputs are the components $(v_1, v_2, v_3)$ of a velocity vector, and, at the next layer, these have been multiplied by weights, summed, subtracted from a threshold, and passed into a nonlinear function (such as a sigmoid). Given that a nonlinearity is in play, implicitly the output of the multiplication by weights and summation is a dimensionless scalar—it is inconsistent with the passive symmetries of $O(3)$ covariance and units covariance for a nonlinearity to be applied to anything other than a dimensionless scalar. This condition will only be met if implicitly the three weights $(W_1, W_2, W_3)$ that multiply these vector components are themselves the components of an $O(3)$-covariant vector with units of inverse velocity. If the network has any nodes in the next layer that have graph connections (weights) to only one or two of the three components $(v_1, v_2, v_3)$, that is, if the network is sparse in the wrong ways, these implicit conditions cannot be met. The passive symmetries thus also put conditions on network architecture or graph structure.

$L_1$ and $L_\infty$ norms are often inconsistent with the passive symmetries. This is because the sum of absolute values of input components, and the maximum of inputs, are rarely either geometrically, or from a units perspective, covariant. There is a rare exception if all features have the same units, and none of the features are components of geometric objects (they are all dimensionless scalars). Similarly, regularizers favoring flat loss minima (Hochreiter & Schmidhuber, 1997; Dinh et al., 2017; Petzka et al., 2021) are often not units covariant, changing their values under certain weight transformations that leave the overall function invariant. If reformulated as a regularizer that is a covariant function of the training points, this problem vanishes (von Luxburg et al., 2004). Data normalization, batch normalization, and layer normalization are all generally brutal and often violate model equivariances (Aalto et al. 2022 and see Section 9).

Finally, we mention that passive symmetries play a crucial role also when it comes to latent variable models, since unobserved latent factors usually come with a large class of allowed gauge transformations (permutations, rotations in the latent space, and coordinate-wise nonlinear transformations) which should be incorporated correctly when studying notions of identifiability (Khemakhem et al., 2020; Buchholz et al., 2022).

## 9    Example: Normalization

To make contemporary neural network models numerically stable, it is conventional to normalize the input data, and possibly also layers of the network with either layer normalization or batch normalization. This normalization usually involves shifting and scaling the features or latent variables to bring them closer to being zero mean and unit variance (or something akin to these).

For our purposes here, let's focus on data normalization and assume that it works as follows: The training data contains features $X$ which are $N \times M$, where $N$ is the number of training data examples, and $M$ is the number of features per data point. In the simplest possible form, the training data $X$ are given a shift and scaling as

$$X' \leftarrow \sigma^{-1}\left(X - \mu\right), \tag{4}$$

where $\sigma$ is some scale or list of $M$ scales derived from the features in the training data set, and $\mu$ is some shift or list of $M$ shifts derived from the features in $X$. It is not uncommon for $\sigma$ to be a root-variance of the features (or a mean absolute deviation or other distribution-width measure) and for $\mu$ to be a mean or median.

Naïve normalization like this will in general break the passive symmetries of geometry and units (Aalto et al., 2022). For one, different individual features in $X$ will have different units. It does not make sense to add or average (nor add nor average in the square) features with different units. For another, if a subset of 3

features in $X$ are the components of a 3-vector subject to $O(3)$ symmetry or a subset of 9 features in $X$ are the components of a tensor subject to $O(3)$ symmetry, these components cannot be summed or medianed without violating $O(3)$ equivariance (and thus passive-symmetry covariance), nor can they be independently scaled without violating $O(3)$.

What should be done instead? For one, the elements of the training data $X$ that correspond to 3-vector components cannot be all treated monolithically in the computation of the shift $\mu$ and scale $\sigma$. Instead, the vectors must be dotted into themselves each individually to construct scalar norms. Those scalar norms can subsequently be summed or medianed or averaged or square-rooted. Usefully, the root-mean-square of the components of a 3-vector can be seen as the square root of $1/3$ of the dot product of the vector with itself. At application time, the three components of any 3-vector must be scaled identically, not independently, otherwise the vector is (in general) being arbitrarily rotated. In this 3-vector case, the shift $\mu$ cannot be a single number but instead, the shift must itself be a 3-vector which is an $O(3)$-equivariant function of the input vectors. That's natural in many but not all normalization methods in use at present.

For another, the features in $X$ that correspond to the elements of tensors must not have their components treated equally in the computation of the shift $\mu$ and scale $\sigma$. Instead, the tensors must have their spectral norms taken (or have some other $O(3)$-equivariant norms must be taken), with each tensor treated individually. Those norms can subsequently be summed or medianed or averaged or square-rooted. Unfortunately, taking spectral norms is more expensive than taking moments. Once again, at application time, the tensor components cannot be independently scaled with nine different scales; instead, one scale must be applied consistently to all nine components. Once again, the shift applied cannot be a single number applied to all components but instead the shift must itself be a tensor. These vector and tensor considerations suggest that data normalization needs to be substantially modified to achieve covariance, that is, to accommodate the passive symmetries of vectors and tensors.

For yet another, elements of $X$ with different units must be treated differently. For shifts, it makes sense to subtract from each quantity a shift $\mu$ that is computed from only those features that share units with the feature in question. For scales, it might make sense to find base-unit scales that make the range of scaled features as close as possible to having unit variance.

## 10    Discussion

In this conceptual contribution, we argue that passive symmetries are in play in essentially all ML or data-analysis tasks. They are exact, and true by definition, since they emerge from the redundancies or freedom in coordinate systems, units, or data representation. Enforcement of these symmetries should improve enormously the generalization capabilities of ML methods. We demonstrate this with toy examples.

In practice, implementation of the passive symmetries in an ML problem might be very difficult. One reason is that the symmetries are only exact when all relevant problem parameters (including often fundamental, unvaried constants) are known and included in the learning problem. If the problem has a passive symmetry by a group $G$, but there are missing elements $K$ in the problem formulation (such as Planck's constant or the gravity vector in Section 5), then the active symmetry that is actually in play is the subgroup $H$ of $G$ that fixes $K$. Naively there should be no difference in the in-distribution performance between enforcing the symmetry by $H$, or including $K$ to the inputs and enforcing the symmetry induced by $G$. However, using the full group equivariance is conceptually more elegant and it allows for out-of-distribution generalization (the model can generalize to settings where $K$ has changed). Of course, these unknown constants or features $K$ are pieces of essential contextual information and can be difficult to find or learn. In our toy examples, we show that with sufficient knowledge of the problem (rich training data and knowledge of the group of passive symmetries) the relevant constant $K$ can be learned from the data, including the Planck constant (for the blackbody-radiation problem) and the gravitational acceleration vector (for the double-pendulum example). Identifiability issues may arise when more constants or non-constant features are missing.

Another difficulty is that some kinds of symmetries are hard to enforce. For example, complete coordinate diffeomorphisms and problem reparameterizations involve enormous groups which are hard to implement in any realistic ML method. That said, many groups have been implemented usefully, including translations,

rotations, permutations, changes of units, and some coordinate transformations (see Weiler et al. 2021a for a review of the latter).

In addition to the exact (and true by definition) passive symmetries, and the observed active symmetries, there are other kinds of approximate or weakly broken symmetries we might call *observer symmetries*. These arise from the point that the content of a data record (an image, say) is independent of the minor choices made by the observer in taking that data record (shooting the image, say). The details of the six-axis location and orientation of the camera, and of the exposure time and focus, can be changed without changing the semantic or label content of the image. These symmetries are approximate, because these changes don't lead to invertible changes in the recorded data; there is no group or groupoid in the space of the data. However, the success of convolutional structure in image models might have to do with the importance of these observer symmetries. There is much more to do in this space.

One topic that we did not address in this note is the use of symmetries as design principle or *inductive bias* and their interaction with optimization. One feature of modern machine learning is that the models used to fit the data are often overparameterized (Zhang et al., 2021). That means that the model has more parameters than training points, sometimes several orders of magnitude more. In particular, many possible functions fit the data perfectly, but not all of them generalize well to new data points (Hogg & Villar, 2021) (see the double descent, benign overfitting, and related ideas Belkin et al. 2019; Bartlett et al. 2020; Nakkiran et al. 2021; Huang et al. 2020; Sonthalia et al. 2023, among many others). Therefore, the design of the class of functions is very important for the performance of the models. Through the years, the classes of functions that have been empirically proven to be successful, incorporate symmetries in their design (for example, convolutional neural networks, graph neural networks, transformers, etc). It is not fully understood why these models are successful and whether the symmetries are a key aspect for their success.

**Acknowledgement:** It is a pleasure to thank Roger Blandford (Stanford), Ben Blum-Smith (JHU), Wilson Gregory (JHU), Nasim Rahaman (MPI-IS), and Shubhendu Trivedi for valuable comments and discussions. This project was started at the meeting *Machine Learning for Science* at Schloss Dagstuhl, 2022 September 18–23. SV was partially supported by ONR N00014-22-1-2126, the NSF–Simons Research Collaboration on the Mathematical and Scientific Foundations of Deep Learning (MoDL) (NSF DMS 2031985), NSF CISE 2212457, NSF CAREER 2339682, and an AI2AI Amazon research award. This project made use of open-source software, including Python, jax, objax, jupyter, numpy, matplotlib, scikit-learn. All code used in this project is available at repositories `https://github.com/weichiyao/TowardsFullyCovariantML`.

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

# A    Glossary

We provide here a glossary to translate terminologies among machine learning, mathematics, and physics.

**active symmetry:** A symmetry is *active* when it is an observed or empirical regularity of the laws of physics. Examples include the observation that the fundamental laws don't depend on the location or time at which the experiment takes place. We provide a formal definition in Section 4.

**conservation law:** We say that a quantity obeys a *conservation law* if changes in that quantity (with time) inside some closed volume can are quantitatively explained by fluxes of that quantity through the surface of that volume. Active symmetries can lead to conservation laws in dynamical systems (when the dynamics is Lagrangian; Noether 1918).

**coordinate freedom:** When physical quantities are measured, or represented in a computer, they must be expressed in some coordinate system. The redundancy of this representation—the fact that the investigator had many choices for the coordinate system—leads to the passive symmetry *coordinate freedom*: If the inputs to a physics problem are moved to a different coordinate system (because of a change in the origin or orientation), the outputs of the problem must be correspondingly moved. In much of the literature "coordinate freedom" is only used in relationship to general covariance, but it applies in all contexts (including non-physics contexts) in which a coordinate system has been chosen.

**covariance:** When a physical law is equivariant with respect to a passive symmetry, then the law is said to be *covariant* with respect to that symmetry.

**dimensions:** Dimensions are the abstract generalization of units. Two quantities that can be given the same units (possibly with a units change for one of them) have identical *dimensions*.

**dimensional analysis:** The technique in physics of deducing scalings by consideration of units covariance is *dimensional analysis*.

**equivariance:** Let $G$ be a group that acts on vector spaces $X$ and $Y$ as $\rho_X : G \to \mathrm{Sym}(X)$ and $\rho_Y : G \to \mathrm{Sym}(Y)$ respectively. Namely, $\rho_X$ (and similarly $\rho_Y$) maps each element of $G$ to a bijective function from $X$ to itself satisfying some rules described in the definition of "group action". We say that a function $f : X \to Y$ is *equivariant* if for any group element $g \in G$ and any possible input $x$, the function obeys $f(\rho_X(g)(x)) = \rho_Y(g)(f(x))$. This is typically expressed by saying that the following commutative diagram commutes for all $g \in G$:

$$
\begin{array}{ccc}
X & \xrightarrow{\ f\ } & Y \\
{\scriptstyle \rho_X(g)}\big\downarrow & & \big\downarrow{\scriptstyle \rho_Y(g)} \\
X & \xrightarrow{\ f\ } & Y
\end{array}
\tag{5}
$$

The actions of $G$ in $X$ and $Y$ induce an action on the space of maps from $X$ to $Y$. If $f \in \mathrm{Maps}(X, Y)$ then we can define $\rho_{XY} : G \to \mathrm{Sym}(\mathrm{Maps}(X, Y))$ such that $\rho_{XY}(g)(f) = \rho_Y(g) \circ f \circ \rho_X(g)^{-1}$. The equivariant maps are the fixed points of this action.

While an equivariance is a mathematical property of a map, in this contribution we use the word "equivariance" mainly in the context of active symmetries, and "covariance" in the context of passive symmetries.

**gauge freedom:** Some physical quantities in field theories (for example the vector potential in electromagnetism) have additional degrees of freedom that go beyond the choice of coordinate system and units. These freedoms lead to additional passive symmetries that are known as *gauge freedom*.

**general covariance:** The covariance of relevance in general relativity (Einstein, 1916) is known as *general covariance*. Because general relativity is a metric theory on an intrinsically curved spacetime of $3 + 1$ dimensions that is invariant to arbitrary diffeomorphisms of the coordinate system, this is a very strong symmetry. General covariance is sometimes called "coordinate freedom", but it is a special case thereof.

**group:** A *group G* is a set with a binary operation $\cdot$ satisfying the following: *(1)* Associativity: for all $a, b, c \in G$ we have $(a \cdot b) \cdot c = a \cdot (b \cdot c)$; *(2)* Identity element: there exists an element $e \in G$ so that $e \cdot a = a \cdot e = a$ for all $a \in G$; *(3)* Inverse: for all $a \in G$ there exists a unique element $b$ such that $a \cdot b = b \cdot a = e$; this element is the inverse of $a$ and it is typically denoted as $a^{-1}$.

**group action:** We consider a group $G$, a space $X$, and a map $\rho : G \to \mathrm{Sym}(X)$, where $\mathrm{Sym}(X)$ denotes the bijective maps from $X$ to itself. Lightly abusing notation, sometimes this map is expressed as $\rho : G \times X \to X$. We say $\rho$ is an *action* of $G$ on $X$ if it satisfies the following properties: *(1)* Identity: $\rho(e)(x) = x$ for all $x \in X$, where $e$ is the group identity element. *(2)* Compatibility: $\rho(a)(\rho(b)(x)) = \rho(a \cdot b)(x)$ for all $a, b \in G$ and $x \in X$.

**group representation:** A *representation* of a group $G$ is an action $\rho : G \to \mathrm{GL}(V)$, where $\mathrm{GL}(V)$ is the space of invertible linear transformations of the vector space $V$. Sometimes it is said that $V$ is a representation of $G$ (and the action is implicitly known). When $V$ is a finite dimensional vector space, the group representation allows us to map the multiplication of group elements to multiplication of matrices.

**invariance:** An equivariance in which the action in the output space is trivial is called an *invariance*. Physicists sometimes use the word invariant (gauge invariant, for example) for things we would call covariant.

**passive symmetry:** A symmetry is *passive* when it arises from a choice in the representation of the data. Examples include coordinate freedom, gauge freedom, and units covariance. These symmetries are exact and true by definition. We provide a formal definition in Section 4.

**scalar:** A number (with or without units), whose value does not depend on the coordinate system in which it is represented, is a *scalar*. Thus, say, the charge of a particle is a scalar, but the $x$ coordinate of its velocity is not a scalar.

**symmetry:** Given a mathematical object $X$ of any sort, (such as a manifold, metric space, equation, etc), any intrinsic property of $X$ which causes it to remain invariant under certain classes of transformations (such as rotation, reflection, inversion, or other operations) is called a *symmetry*. For our purposes, the symmetries of interest can be expressed as equivariances or invariances, both defined above.

**tensor:** A multi-linear function of $k - 1$ vectors that outputs a vector, or a multi-linear function of $k$ vectors that outputs a scalar, is a *k-tensor*. A rectangular array of data is not usually a tensor according to this definition. A vector can be seen as a 1-tensor (the linear function corresponding to the vector being the inner product with that vector), and a scalar can be seen as a 0-tensor.

There is an alternative definition of tensor in terms of transformations with respect to the $O(d)$ group, analogous to the primary definition of "vector" below.

**units:** All physical quantities are measured with a system of what we call *units*. A quantity can be transformed from one unit system to another by multiplication with a dimensionless number. Almost all quantities—including almost all scalars, vectors, and tensors—have units.

**units covariance:** The left-hand side and the right-hand side of any equation must have the same units. This symmetry is called (by us) *units covariance* (contra Villar et al. 2023 where it is called "units equivariance").

**vector:** An ordered list of $d$ numbers, all of which have the same units, that is subject to the passive $O(d)$ symmetry corresponding to coordinate-system rotations, is a *vector* in $d$ dimensions. These rotations are performed by the standard rotation (and reflection) matrix representation of the elements of $O(d)$. See the definition of "tensor" for an alternative definition: The inner (or dot) product of two vectors produces a scalar; for this reason, a vector can be seen as a 1-tensor. A generic ordered list of $d$ features is not usually a vector according to this definition.

## B  Units covariant regression: The black body experiment

For our first experiments, we consider the problem of black-body radiation, in which the intensity $B_\lambda(\lambda; T)$ of radiation at wavelength $\lambda$ is a function of temperature $T$. The true relationship is given by (3), and we use it to construct a toy inference problem in which the units of the input quantities are non-trivial.

The training data are made by computing intensities $B_\lambda(\lambda; T)$ according to (3) at five temperatures (300, 1000, 3000, 10000, and 30000 K), and on a wavelength grid of 40 wavelengths on a logarithmically spaced grid from $10^{-8}$ to $10^{-4}$ m. To each evaluation of intensity we add a draw of zero-mean Gaussian noise with root-variance 0.1 times the true value of that intensity. The training data are cut on intensity, such that only data points with intensities above $10^6$ in SI units are retained. The remaining data points so constructed comprise the training data, with features $x$ being the wavelengths and temperatures, and labels $y$ being the noisy values of the intensity. In some cases we augment the features with copies of the speed of light $c$ and the thermal (Boltzmann's) constant $k$. At each of the training data points, there are two or four fundamental features (the first 2, or all, of the list $\lambda, T, c, k$), depending on the regression.

The test data are similarly generated except that they are generated at temperatures of 6000, 20000, and 60000 K. That is, the test data extend outside the temperature range of the training data, and the test and training data have no temperatures in common. Once the intensity cut is made, there are 126 data points in the training set and 108 in the test set.

We perform three regressions, the output of which are shown in Figure 2: In the first (dubbed "standard MLP"), a 3-layer multi-layer perceptron from scikit-learn, with hidden-layer sizes of 20, 20, and 20 is trained on the logarithm of the first two training-set features (features $\lambda, T$) and the logarithm of the training-set labels. The trained MLP is used to predict the test-set labels. This model incorporates no explicit symmetries.

In the second regression (dubbed "units covariant without extra dimensional constant"), the same design of MLP is used, but instead of inputting the logarithms of the fundamental features, the input to the MLP is the set of logarithms of all the dimensionless combinations possible to make from the four features $\lambda, T, c, k$, following the approach of Villar et al. (2023). This approach uses the Smith normal form (Stanley, 2016) to find products of integer powers of the input features that are dimensionless, and one product of integer powers of the inputs that has the same units as the label. The output of the MLP is multiplied by the combination of the input features with the correct units to be an intensity. In this case, there are no non-trivial dimensionless features, so the MLP sets one amplitude only, and the predictions are pure power laws (as is visible in Figure 2).

In the third regression (dubbed "units covariant with extra dimensional constant"), the features are augmented with one dimensional quantity $q$, and then the same procedure is followed as for the previous units-covariant case. If the quantity $q$ has units that are independent (in the appropriate sense) from the units of the four fundamental features, then a new dimensionless constant becomes possible and the regression becomes non-trivial. The units of $q$ are chosen from an exhaustive list of all possible units that involve powers between $-2$ and 2 of kg, m, s, and K. The value of $q$ is chosen, at each of these units choices, to be such that the range of dimensionless features in the training data has unit median. Again the MLP input is logarithms of dimensionless features and the output is multiplied by the Smith normal form returned combination with the correct units to be an intensity. Many choices for the units and value of $q$ lead to good regression accuracy; all of them make $q$ close to some combination of $h, c, k$, necessarily including $h$.

## C  Springy double pendulum

**Setting.**  We consider the dissipationless spherical double pendulum with springs, with a pivot $o$ and two masses connected by springs. The kinetic energy $\mathscr{T}$ and potential energy $\mathscr{U}$ of the system are given by

$$KE = \frac{|\mathbf{p}_1|^2}{2m_1} + \frac{|\mathbf{p}_2|^2}{2m_2}, \tag{6}$$

$$PE = \frac{1}{2}k_1(|\mathbf{q}_1 - \mathbf{q}_o| - l_1)^2 + \frac{1}{2}k_2(|\mathbf{q}_2 - \mathbf{q}_1| - l_2)^2 - m_1\,\mathbf{g}\cdot(\mathbf{q}_1 - \mathbf{q}_o) - m_2\,\mathbf{g}\cdot(\mathbf{q}_2 - \mathbf{q}_o), \tag{7}$$

where $\mathbf{q}_1, \mathbf{p}_1$ are the position and momentum vectors for mass $m_1$, similarly $\mathbf{q}_2, \mathbf{p}_2$ for mass $m_2$, and a position $\mathbf{q}_o$ for the pivot. The springs have scalar spring constants $k_1$, $k_2$, and natural lengths $l_1$, $l_2$. The gravitational acceleration vector is $\mathbf{g}$. In this work, we fix $\mathbf{q}_o$ with values $(0,0,0)$ in base length units and $\mathbf{g}$ with $(0,0,-1)$ in base acceleration units, as well as $(m_1, m_2, k_1, k_2, l_1, l_2)$ set to $(1,1,1,1,1,1)$, where each element of the list has appropriate base units.

The prediction task is to learn the positions and momenta over a set of $T$ later times $t$ given the initializations of the pendulum positions and momenta at $t_0$,

$$\mathbf{z}(t) = (\mathbf{q}_1(t), \mathbf{q}_2(t), \mathbf{p}_1(t), \mathbf{p}_2(t)), \quad t \in \{t_0, t_1, \ldots, t_T\}. \tag{8}$$

The training inputs consist of $N = 500$ different initializations of the pendulum positions and momenta $\{\mathbf{z}^{(i)}(t_0^{(i)})\}_{i=1}^N$, and the labels are the set of positions and momenta $\{\mathbf{z}^{(i)}(t_1^{(i)}), \mathbf{z}^{(i)}(t_2^{(i)}), \ldots, \mathbf{z}^{(i)}(t_T^{(i)})\}_{i=1}^N$ with $T = 5$. The model is evaluated on a test data set with $T = 150$ and $t_0 = 0$.

**Variants.** For the same prediction task, we consider three different $O(3)$-equivariant models, $f_{\mathsf{Known\text{-}g}}$, $f_{\mathsf{Learned\text{-}g}}$ and $f_{\mathsf{No\text{-}g}}$, depending how the gravitational acceleration vector $\mathbf{g}$ is involved.

**Known-g** The model $f_{\mathsf{Known\text{-}g}}$ is a function that predicts the dynamics:

$$\begin{aligned} f_{\mathsf{Known\text{-}g}} : (\mathbb{R}^3)^4 \times \mathbb{R}^3 \times \mathbb{R}^3 \times \mathbb{R} &\to (\mathbb{R}^3)^4 \\ (\mathbf{z}(0), \mathbf{q}_o, \mathbf{g}, \Delta t) &\mapsto \hat{\mathbf{z}}(\Delta t) \end{aligned} \tag{9}$$

where $\mathbf{g}$ is known as $(0, 0, -1)$ in the base acceleration units and used with positions and momenta as input features.

**Learned-g** The model $f_{\mathsf{Learned\text{-}g}}$ is a function that predicts the dynamics:

$$\begin{aligned} f_{\mathsf{Learned\text{-}g}} : (\mathbb{R}^3)^4 \times \mathbb{R}^3 \times \mathbb{R} &\to (\mathbb{R}^3)^4 \\ (\mathbf{z}(0), \mathbf{q}_o, \Delta t) &\mapsto \hat{\mathbf{z}}(\Delta t) \end{aligned} \tag{10}$$

where $\mathbf{g}$ is unknown but set as a learnable variable and used with positions and momenta as input features.

**No-g** The model $f_{\mathsf{No\text{-}g}}$ is a function that predicts the dynamics:

$$\begin{aligned} f_{\mathsf{No\text{-}g}} : (\mathbb{R}^3)^4 \times \mathbb{R}^3 \times \mathbb{R} &\to (\mathbb{R}^3)^4 \\ (\mathbf{z}(0), \mathbf{q}_o, \Delta t) &\mapsto \hat{\mathbf{z}}(\Delta t) \end{aligned} \tag{11}$$

where $\mathbf{g}$ is unknown and not used as an input feature.

We evaluate the performance of the three predictive models based on the state relative error at a given time $t$ in terms of the positions and momenta of the masses,

$$\text{State.RelErr}(t) = \frac{\sqrt{(\hat{\mathbf{z}}(t) - \mathbf{z}(t))^\top (\hat{\mathbf{z}}(t) - \mathbf{z}(t))}}{\sqrt{\hat{\mathbf{z}}(t)^\top \hat{\mathbf{z}}(t)} + \sqrt{\mathbf{z}(t)^\top \mathbf{z}(t)}}, \quad t \in \{t_1, \ldots, t_T\}, \tag{12}$$

where $\hat{\mathbf{z}}(t)$ denotes the predicted positions and momenta at time $t$ and $\mathbf{z}(t)$ the ground truth.

**Model.** In all the variants of the experiment, the learned function $f$ is implemented by a Hamiltonian neural network (Sanchez-Gonzalez et al., 2019). The model learns an scalar $O(3)$-invariant function $\mathscr{H}$ of the input vectors, and uses a symplectic integrator to predict the dynamics:

$$\frac{\mathrm{d}\mathbf{q}_s}{\mathrm{d}t} = \frac{\partial \mathscr{H}}{\partial \mathbf{p}_s}, \qquad \frac{\mathrm{d}\mathbf{p}_s}{\mathrm{d}t} = -\frac{\partial \mathscr{H}}{\partial \mathbf{q}_s}, \qquad s = 1, 2. \tag{13}$$

We implement $\mathscr{H}$ to be $O(3)$-invariant by simply restricting $\mathscr{H}$ to be a function of the inner products of the possible input vectors $\mathbf{q}_1, \mathbf{q}_2, \mathbf{p}_1, \mathbf{p}_1, \mathbf{q}_o$ and possibly $\mathbf{g}$, following the fundamental theorem for invariant

|  | Training MSE | Test MSE | Test rollout error |
|---|---|---|---|
| No normalization | $0.0000 \pm 0.0000$ | $0.0028 \pm 0.0013$ | $0.0029 \pm 0.0001$ |
| Non-equivariant normalization | $0.0184 \pm 0.0006$ | $0.2587 \pm 0.0233$ | $0.1177 \pm 0.0043$ |
| Equivariant normalization | $0.0000 \pm 0.0000$ | $0.0070 \pm 0.0011$ | $0.0053 \pm 0.0007$ |

Table 1: Empirical performance of the different data normalization procedures.

theory for $O(d)$ (see Weyl 1946 for the theory, and Villar et al. 2021 for a discussion on machine learning implications). Namely,

$$\mathscr{H}_{\text{variant}}(\mathbf{q}_1, \mathbf{q}_2, \mathbf{p}_1, \mathbf{p}_2) = h((\mathbf{v}^\top \mathbf{w})_{\mathbf{v}, \mathbf{w} \in \mathscr{V}_{\text{variant}}}) \tag{14}$$

where $\mathscr{V}_{\text{Known-g}} = \{\mathbf{q}_1, \mathbf{q}_2, \mathbf{p}_1, \mathbf{p}_2, \mathbf{g} = (0, 0, 1)\}$; $\mathscr{V}_{\text{Learned-g}} = \{\mathbf{q}_1, \mathbf{q}_2, \mathbf{p}_1, \mathbf{p}_2, \mathbf{g}\}$ where $\mathbf{g}$ is a variable that is jointly optimized with the parameters that define $\mathscr{H}$; $\mathscr{V}_{\text{No-g}} = \{\mathbf{q}_1, \mathbf{q}_2, \mathbf{p}_1, \mathbf{p}_2\}$. Note that $\mathbf{q}_o = 0$ so all the corresponding inner products are zero and therefore dropped from the model. The model predictions are computed as

$$f_{\text{variant}} = \text{symplectic-integration}(\mathscr{H}_{\text{variant}}(\mathbf{q}_1, \mathbf{q}_2, \mathbf{p}_1, \mathbf{p}_2), \Delta_t). \tag{15}$$

Note that this model is time-translation-equivariant, can be $O(3)$-equivariant (depending on the variant), but it isn't units-covariant.

Our experiments show that the $O(3)$-equivariant variant with known gravity has the best performance, closely followed by the model with learned gravity. We note that the model learns a vector very close to the ground-truth gravity vector, and therefore learns an equivariant model. The non equivariant model with no gravity vector performs quite poorly, as expected. The results are presented in Figure 2.

**Data normalization** For the data normalization experiment, we consider the standard variant where the gravity vector is known. We normalize the inputs of $\mathscr{H}$ in (15) before computing the inner products. We consider the input data matrix with rows indexed by $i = 1, \ldots, N$ and columns of the form

$$(\mathbf{q}_1^{(i)}(t), \mathbf{q}_2^{(i)}(t), \mathbf{p}_1^{(i)}(t), \mathbf{p}_2^{(i)}(t), \mathbf{g} = (0, 0, -1))_{t=t_0, \ldots, t_{150}} \tag{16}$$

**Non-equivariant data normalization** For the naive data normalization we consider the data matrix with $N$ rows and $3 \times 4 \times 150$ columns (the first 4 columns of (16)) and we normalize it by columns. Namely, for each entry of the matrix we subtracts the mean of its column and divide by the standard deviation of the column. This operation does not respect any of the symmetries of the problem, therefore when we feed this normalized data to our standard model (15), the performance is bad, as shown in Table 1.

**Equivariant data normalization** The equivariant data normalization takes all the vectors $\{\mathbf{q}_1^{(i)}(t), \mathbf{q}_2^{(i)}(t)\}_{i,t}$ and computes their $3 \times 3$ empirical covariance matrix $C_{\mathbf{q}}$. Similarly it computes $C_{\mathbf{p}}$. The normalized model uses the formulation (15) with

$$\mathscr{V}_{\text{normalized}} = \left\{ \frac{\mathbf{q}_1}{\sqrt{(1/3)\operatorname{tr}(C_{\mathbf{q}})}}, \frac{\mathbf{q}_2}{\sqrt{1/3\operatorname{tr}(C_{\mathbf{q}})}}, \frac{\mathbf{p}_1}{\sqrt{(1/3)\operatorname{tr}(C_{\mathbf{p}})}}, \frac{\mathbf{p}_2}{\sqrt{(1/3)\operatorname{tr}(C_{\mathbf{p}})}}, \mathbf{g} = (0, 0, -1) \right\}. \tag{17}$$

The performance is comparable to the one with no normalization, and several orders of margnitude better than the performance of the non-equivariant data normalization method.

