# OpenReview forum: "Towards fully covariant machine learning"
_TMLR — Accepted by TMLR_

### Review · Reviewer_JES3 · 2023-09-25

**Summary Of Contributions:**

This paper is a think-piece that argues that machine learning models should encode passive symmetries (i.e. natural equivariances due to the arbitrariness of the mathematical system used to encode phenomena). The paper first introduces notions of passive versus active symmetries, and discusses how much of the work in invariant/equivariant machine learning has targeted active symmetries but largely ignored passive symmetries. The authors use toy experiments to demonstrate that taking passive symmetries into account can improve data efficiency. The authors then comment on connections between passive symmetry and causal learning, and describe how our choices of nonlinear activation function and normalization layers lead to violations of passive symmetry.

**Audience:**

Yes

**Claims And Evidence:**

Yes

**Requested Changes:**

## Major (will not vote for acceptance without these changes)

- Add Section 6 experimental details to the appendix

## Very nice to have (but not absolutely necessary)

- Can you include experiments that demonstrate how using nonlinearities/normalization that violate passive symmetries actually impacts performance? This section would become much stronger (and easier to understand) with some explicit experimental examples.
- Shorten the paper and/or move some of the sections into supplementary material - ideally to fit within 12 pages

## Extremely minor (typos)

- Page 6 towards the bottom "sometimes Einstein summation notation, Einstein 1916" <- should this be "sometimes *called* Einstein summation notation"?
- Page 8 "The two approaches are theoretically eqiuvalent" <- should be "three" approaches

**Strengths And Weaknesses:**

## Strengths

Overall, I found this paper to be very thought provoking and interesting to read. While invariance/equivariances have become quite popular in machine learning models, this paper demonstrates a very notable area of opportunity (passive symmetries) that have been largely ignored by the existing literature. Though this paper is more of a think piece rather than a traditional ML publication, I believe that the ideas it conveys will be of interest not only to those who study symmetry in ML but also to the larger ML community as a whole.

**Clarity.** Overall the paper is very well written and easy to follow, even for someone who does not have a strong background in group theory. However, there is a bit of redudancy that could make the paper more streamlined (see Weaknesses).

**Experiments.** The toy settings in Section 6 convey a promising illustration of the authors' main ideas.


## Weaknesses

- The experiments in Section 6 assume significant familiarity with prior work. The authors should include a more thorough description of the different methods, perhaps in the supplementary material, to assist those not familiar with the work.
- Sections 9 and 10 felt a bit too speculative, especially after the toy experiments in Section 6 provide nice grounded evidence. It would be nice to include some specific toy experiments here to demonstrate the consequences of violating passive symmetries. Right now, these sections simply demonstrate a violation of principles.
- The paper feels unnecessarily long, and (at least in the first 6 sections) somewhat redundant/repetative. I think that this paper could fit within the standard 12 page limits of a TMLR submission without needing to cut much content. For example:
  - Get rid of the page break after the abstract
  - Move the glossary to the appendix (it is very helpful, but disrupts the reading flow).
  - Sections 3-5 seem to repeat many of the same ideas but with slight variation. It is possible that including the formal definitions of Section 5 before introducing the content in Section 3 could reduce the need to re-introduce passive and active symmetries multiple times.

---

> ### Author Response · Authors · 2023-10-11
>
> Suggested changes:
>
> - The authors should include a more thorough description of the different methods, perhaps in the supplementary material, to assist those not familiar with the work.
>
> We will add a description of the different methods in the literature and experimental results in an appendix.
>
> - It would be nice to include some specific toy experiments here to demonstrate the consequences of violating passive symmetries. Right now, these sections simply demonstrate a violation of principles.
>
> We are adding a new experiment (see general comment)
>
> - The paper feels unnecessarily long, and (at least in the first 6 sections) somewhat redundant/repetative. I think that this paper could fit within the standard 12 page limits of a TMLR submission without needing to cut much content. For example: Get rid of the page break after the abstract / Move the glossary to the appendix (it is very helpful, but disrupts the reading flow) / Sections 3-5 seem to repeat many of the same ideas but with slight variation. It is possible that including the formal definitions of Section 5 before introducing the content in Section 3 could reduce the need to re-introduce passive and active symmetries multiple times.
>
> We will take all of these suggestions; these are all useful and reasonable. We agree that the paper can fit in the 12 pages with some reasonable changes.
>
> - Add Section 6 experimental details to the appendix
>
> We will write two new appendices, one on the experiment that’s there now, and one on the new experiment (described above).
>
> - Can you include experiments that demonstrate how using nonlinearities/normalization that violate passive symmetries actually impacts performance? This section would become much stronger (and easier to understand) with some explicit experimental examples.
>
> This is the main change we are implementing (see above).
>
> - Shorten the paper and/or move some of the sections into supplementary material - ideally to fit within 12 pages
>
> See above.
>
> - Page 6 towards the bottom "sometimes Einstein summation notation, Einstein 1916" <- should this be "sometimes called Einstein summation notation"?
> - Page 8 "The two approaches are theoretically eqiuvalent" <- should be "three" approaches
>
> Will fix these.

---

> > ### Comment · Reviewer_JES3 · 2023-10-12
> >
> > Thank you for your response. I look forward to seeing an updated manuscript. Overall I believe this work will be thought provoking and of value to the community.

---

### Review · Reviewer_VRS3 · 2023-09-27

**Summary Of Contributions:**

The paper discusses ways of incorporating passive symmetries into ML frameworks at a conceptual level. This is different than the study of active symmetries as done in equivariant network literature.

**Audience:**

No

**Claims And Evidence:**

No

**Requested Changes:**

TMLR may not be best suited for such conceptual contributions so I'd like to recommend submitting elsewhere.

**Strengths And Weaknesses:**

The paper is written at a conceptual level. The major weakness is the lack of concrete results (theory or experiments) which can be evaluated in the reviews.

In the abstract, the paper arguably translates between different languages of physics, mathematics, and machine learning. Although I see such traces in the glossary, these vocabularies are not fundamental in the later discussion.

---

> ### Author Response · Authors · 2023-10-11
>
> Requested changes:
>
> - The paper is written at a conceptual level. The major weakness is the lack of concrete results (theory or experiments) which can be evaluated in the reviews.
>
> We will add an extra numerical experiment (see above).
>
> - In the abstract, the paper arguably translates between different languages of physics, mathematics, and machine learning. Although I see such traces in the glossary, these vocabularies are not fundamental in the later discussion.
>
> This point is fair; we will reduce the emphasis on this point in the abstract, move the glossary to the appendix, and take care to make translations in the text where they are useful.

---

### Review · Reviewer_xD5g · 2023-09-27

**Summary Of Contributions:**

The paper considers the problem of building predictive models that are equivariant with respect to passive symmetries, i.e. symmetries that arise from arbitrary choices such as coordinate systems. These are distinct from so called "active" symmetries which are (often approximate) empirically verified symmetries of systems. The paper contains an extensive introduction to the main concepts, some validation of a proposed strategy on two toy examples, and some dos and donts recommendations for practitioners.

**Audience:**

Yes

**Broader Impact Concerns:**

No ethical concerns here.

**Claims And Evidence:**

Yes

**Requested Changes:**

- The procedure followed in the practical examples should be explained in detail, possibly providing pseudocode or access to notebooks.
- A clearer explanation of the novelty (I guess the introduction of an additional feature vector to account for missing information which is learnt from data) should be given.
- The method of Villar et al which is at the root of the proposed method should also be described, at least summarily, in order to make it clearer what are the main changes.

**Strengths And Weaknesses:**

Strengths:
- The paper is clear and well written;
- The distinction between active and passive symmetries is potentially important and not widely appreciated in the community;
- the historical perspective and links with physics and dimensional analysis are interesting and illuminating.

Weaknesses:
- The paper is somewhat philosophical in outlook. The authors are clearly aware of this and per se it's not a problem, but I think more effort should be made to explain the practical implications.
- The main contribution is not entirely clear (see below).

---

> ### Author Response · Authors · 2023-10-11
>
> Requested changes:
>
> - The procedure followed in the practical examples should be explained in detail, possibly providing pseudocode or access to notebooks.
>
> We will publish all the notebooks in a single github repository, so that they are very easy to run.
>
> - A clearer explanation of the novelty (I guess the introduction of an additional feature vector to account for missing information which is learned from data) should be given.
>
> To the list of contributions we give in the Introduction, we will add a specific bullet point about the circumstances in which an additional learned feature converts a covariant problem into an actively equivariant problem.
>
> - The method of Villar et al which is at the root of the proposed method should also be described, at least summarily, in order to make it clearer what are the main changes.
>
> We will add a description of the experimental setup in the appendix.

---

### Author Response · Authors · 2023-10-11

**Comment to all reviewers**
We appreciate the reviews. They are constructive and we agree with all the proposed changes. We will change the manuscript to incorporate the feedback and add additional code and experiments. We’d like to request some time to do this (we feel like 6 weeks would be a reasonable time frame if it works for TMLR).

*New experiment: covariance in data normalization.*
As two of the referees requested additional experiments, we will perform the following experiment. We consider the problem of predicting the dynamics of a double pendulum with springs. Each input data point input consists of 3-dimensional positions and velocities of two masses connected by springs at times $t=0,...,T-1$, and the prediction consists of the positions and velocities of the masses at times $t=T,…, T+k$.
The input data can be represented as a tabular data $X$ of dimension $NT\times 12$ (the initial conditions, which consist of two positions and two velocities).
We consider the following forms of data normalization:

X <- X - mean(X)

X <- std(X)$^{-1}$ X

Where mean(X) could be any average-like quantity over the columns of the training set, or the rows of the training set. Similarly std(X) could be any dispersion-like quantity. Our experiment (we expect) will show that all of these normalizations will break the covariance of the problem and hurt performance. We will show that replacing the mean(X) and std(X) operations with equivariant means and dispersions (which respect O(d) symmetry for the vectors) and separating the position and velocity normalizations (because they have different units) restores covariance and improves training and performance.

We propose that we will upload a new version of the manuscript with the suggested changes within 6 weeks.

---

> ### Comment · Action_Editors · 2023-10-27
> **On the future update of the paper**
>
> I would like to thank all reviewers for their very constructive feedback, which mostly align. Given the discussions I believe it makes sense to allow for the 6 weeks required by the authors for an update. The final decision will be taken at this time accordingly.

---

### Decision · Action_Editor_7QSq · 2023-12-27

**Recommendation:** Accept as is

**Comment:**

The paper adresses timely questions about symmetries in ML models and how taking "passive symmetries" into account may affect performance. The paper's scope is quite general and may catch the attention of a broad audience. It provides detailed numerical experiments backing up the proposed ideas and conjectures. The experiments are convincing and sufficiently detailed to be reproduced. However, as a general rule I recommend the authors to add a link to the code before the final submission.

**Audience:**

All authors agree that this non-standard paper, both in its subject and presentation, are thought provoking and stimulating for a broad audience. Both theorists and practitioners will find some interesting and novel ideas to exploit here.

**Claims And Evidence:**

Given that the paper is rather high-level, the referees first expressed some concerns related to the lack of sufficient numerical experiments and/or precise statements. A major revision has been provided which adresses these main concerns. In particular, new experiments with detailed explanations have been added. However, I did not find the promised link to the codes to run the experiments, I think it would be important to add it before publication.